# Trends in Diagnosis and Surgical Treatment of Bone Metastases among Orthopedic Surgeons

**DOI:** 10.3390/jcm11154284

**Published:** 2022-07-23

**Authors:** Dawid Ciechanowicz, Daniel Kotrych, Filip Dąbrowski, Tomasz Mazurek

**Affiliations:** 1Department of Orthopaedics, Traumatology and Musculoskeletal Oncology, Pomeranian Medical University, 71-281 Szczecin, Poland; 2Department of Orthopaedics, Traumatology and Orthopaedic Oncology, Collegium Medicum University in Zielona Gora, 65-046 Zielona Gora, Poland; daniel.kotrych@pum.edu.pl; 3Department of Orthopaedics and Traumatology, Faculty of Medicine, Medical University of Gdansk, 80-803 Gdansk, Poland; dabrowskif@gmail.com (F.D.); mazurek@gumed.edu.pl (T.M.)

**Keywords:** bone metastases, bone tumor, trends in treatment, diagnosis, survey, orthopedic oncology

## Abstract

Background: The proper diagnosis and treatment of bone metastases are essential for patient survival. However, several strategies for the treatment are practiced. Therefore, the aim of the study was to analyze what factors influence the choice of a method of treatment. Methods: An online survey was conducted within the Polish Society of Orthopedics and Traumatology. It consisted of 45 questions and was divided into four main parts: Participant Characteristics, Diagnosis and Qualification, Treatment, and Clinical Cases. Results: A total number of 104 responses were collected. The most frequently chosen methods were: Intramedullary nail (IMN) + Resection + Polymethyl methacrylate (PMMA) (30.47%) and IMN without tumor resection (42.13%), and in third place, modular endoprosthesis (17.25%). The less experienced group of orthopedic surgeons more often (47.5% vs. 39.5%) decided to perform IMN without tumor resection than the more experienced group (*p* = 0.046). Surgeons from district hospitals less frequently (13.7% vs. 23.1%) would decide to use modular endoprosthesis than surgeons from university hospitals (*p* = 0.000076). Orthopedists who performed ≥ 11 bone metastases surgeries per year would more often use modular endoprosthesis (34.8% vs. 13.2%) than those who performed ≤ 10 operations per year (*p* = 0.000114). Conclusion: Experience, place of work, and the number of metastasis surgeries performed during a year may influence the choice of treatment method in patients with bone metastases.

## 1. Introduction

Cancer is one of the most common causes of death in the world. Every year, approximately 18 million new patients are diagnosed with malignant neoplasms (in Poland, about 168,000 new cases were diagnosed in 2018) [1,2]. However, it is estimated that the number of new cases will continue to rise. A WHO report estimates that in 2030 approximately 22.2 million people worldwide will develop oncological disease [1]. It is worth noting that in approximately 50–70% of these cases, metastatic disease will occur, including metastatic bone disease (Metastatic Bone Disease—MBD) [3,4]. Although the main forms of therapy for patients with bone metastases currently remain chemotherapy and radiotherapy, it is estimated that approximately 20% of all MBD cases require orthopedic treatment [5,6].

Bone metastases are most often observed in the following neoplasms: prostate cancer (85%), breast cancer (70%), lung cancer (40%), and kidney cancer (40%), of which, due to the frequency of occurrence in the population, prostate, breast, and lung cancer account for 80% of all bone metastases [4,7,8]. MBD is most often diagnosed in the spine (87%) and ribs (77%) [4]. However, more than half of the cases (53–63%) develop metastases in the appendicular skeleton. The most common locations are the humerus, femur, and pelvis [4]. This distribution of bone metastases is most likely due to the presence of bone marrow in these locations, which provides a favorable growth-promoting environment for cancer cells [9].

Continuous development in the treatment of patients with MBD can be observed. In the past, the presence of bone metastasis was one of the contraindications for limb-sparing surgery [7]. Currently, among the methods of surgical treatment, there are many reconstruction techniques that can fill even large bone defects, including Modular Endoprostheses or Custom-made Endoprostheses [8,10]. Additionally, the literature shows that patients with a single bone metastasis should be treated radically, similar to patients with primary bone tumors, rather than receive palliative treatment only [6]. This is seen especially in the case of metastases of clear cell carcinoma of the kidney, where the longer survival of patients undergoing radical treatment has been achieved [11,12]. Nevertheless, each case of a patient with MBD should be evaluated individually within the group of oncologists, radiotherapists, and orthopedists. The potential benefits to the patient from surgery should always be considered. Therefore, the orthopedic surgeon should take into account a range of factors, including the patient’s life expectancy, the patient’s general condition, the number of metastases, the advancement of the neoplastic disease, its sensitivity to treatment, the presence of a pathological fracture, and the patient’s quality of life [5,8]. Additionally, the lack of clear guidelines for patients with MBD often makes the qualification process neither easy nor obvious. It is also worth emphasizing that patients with metastases in the appendicular skeleton are not only treated in Orthopedic Oncology Centers. A patient with a pathological fracture of the femur or humerus can be admitted to any other hospital.

Therefore, in order to analyze current trends in the treatment of patients with MBD, we conducted a survey among members of the Polish Society of Orthopedics and Traumatology. Its main goal was to gain information on the most frequently used tools and methods in the diagnosis and qualification of patients with MBD for treatment. We additionally wanted to find an area that could be improved to provide patients with bone metastases with the best possible treatment in the future. Through the survey, we wanted to assess whether the experience, workplace, and number of surgeries performed on patients with MBD had an impact on treatment preferences. We also wanted to confirm which methods of treatment, based on specific clinical cases, are preferred by the participants in the study.

## 2. Materials and Methods

An online-based anonymous voluntary survey was conducted within the Polish Society of Orthopedics and Traumatology (PTOiTR). Between 15 February 2021 and 31 March 2021, a link to a survey was sent in the form of an email with an invitation to participate in the study and a description of the objectives of the study to the members of the PTOiTR. Four emails were sent inviting participation in the study (15 February, 1 March, 15 March, 29 March). The information reached 4823 members. The survey was created and uploaded on the Google online survey platform (Google Forms^®^). For the study, we designed a questionnaire in Polish, as well as an English translation, which are appended as Appendix A. The data from the survey were collected into a separate file in Microsoft Excel 2019 (Microsoft Corporation, Redmond, WA, USA). The accuracy of the data entered with the survey responses was checked by 3 of the study’s authors (DC, DK, FD). The questionnaires containing answers to all the questions were qualified for the study (except the compulsory question 45). The survey consisted of 45 questions and was divided into 4 main parts: Participant Characteristics (questions 1–6), Diagnosis and Qualification for Surgery (7–24), Treatment (25–27), and Clinical Cases (28–44).

The first part of the survey focused on determining the profile of the study group (gender, experience, place of work, number of bone metastasis surgeries performed per year, main area of interest, and confidence in the oncology of the musculoskeletal system). The aim of this part was to determine whether a diverse study group had been obtained and then to check whether factors such as work experience, workplace, and the number of bone metastasis surgeries performed had an impact on the responses in the next part of the questionnaire.

The second part focuses on aspects of the diagnosis of bone metastases and the qualification of patients for surgical treatment (management of a patient with suspected MBD, diagnostic imaging, biopsy, scales used to qualify patients). In addition to the suggested answers, the participants had the opportunity to add their own. Questions 13–24 were based on the Likert scale, in which the respondents could choose the importance of a symptom/factor while qualifying a patient for surgery (1, completely unimportant; 2, slightly unimportant; 3, slightly important; 4, moderately important; 5, highly important). The factors most frequently mentioned in the literature as important in qualifying patients with MBD for surgery were selected [5,8,11].

The treatment section examined how often (on the Likert scale) the respondents use pre-operative embolization for patients with MBD. Furthermore, where the patient is most often referred after the completed surgical treatment of a metastatic tumor was also checked. The last question asked for the preferred form of filling in bone defects after resection of a metastatic lesion. For all the questions in this section, the participants had the opportunity to add their own expanded answers.

In the clinical cases section, 16 theoretical scenarios were presented, containing a short patient characteristic with radiological images of bone metastases. In each clinical case, the participants could choose their preferred method of surgical treatment from among the suggested distractors (single-choice questions). There were 5 different distractors to choose from: Intramedullary nail with tumor resection and using Polymethyl methacrylate (IMN + Resection + PMMA); Intramedullary nail without tumor resection (IMN); Plate-screw fixation device; Modular endoprosthesis with tumor resection; No indications for surgical treatment. However, in each case, the participants had the opportunity to add their own preferred method of treatment if they felt that there was no satisfactory suggestion among the distractors. All the cases differed from each other in such information as the estimated chance of survival, type of primary tumor, presence of pathological fracture, and location of metastasis. All of the cases contained information on the patient’s severe pain, which is one of the main indications for surgical treatment [8]. For information on the estimated life expectancy, the periods <6 months and >12 months were used. Studies suggest the use of surgical treatment with an estimated life expectancy of 3–6 months; therefore, our study examined how the respondents in their practice deal with patients with such a life expectancy [11]. Additionally, the data in the literature indicate the need for more radical surgical treatment among patients with a longer life expectancy (>12 months) [5]. The second information on the patient—the type of cancer—included two histopathological types: breast cancer, which responds well to radiation therapy, and kidney cancer, which shows a moderate response to radiotherapy and shows good treatment results after the radical excision of the metastatic lesion [5,12]. The third piece of information—the presence of a pathological fracture—contains two different items of data: clear pathological fracture and a high risk of pathological fracture (impendic fracture). To be sure that the radiological tests are not misleading, the description clearly indicates which case we are dealing with. The last factor is the location of the metastasis—the femur and the humerus–which was aimed at assessing surgical treatment trends within two different anatomical locations, not only within one bone. One additional question (44) was added to this part of the questionnaire, containing the case of a patient who should be classified according to Capmann’s classification for radical surgical treatment MBD [7]. In the last question (45), the respondents could add any comments regarding the questionnaire and the surgical treatment of MBD in the form of an open question.

Ethical approval was obtained from the Independent Bioethics Committee at the Medical University of Gdańsk (Approval number: NKBBN/746/2021). In email messages, the survey participants were assured that all of the data would be used only for research purposes and that the data set would not be made available to the public. The participants’ answers were anonymous and confidential according to Google’s privacy policy. The participants did not have to give their names or any contact information. In addition, they could stop participating in the study and could leave the questionnaire at any stage before the submission process, and their responses were not saved. A response was saved only by clicking on the “submit” button. By completing the survey, participants acknowledged the above approval form and were consenting to participate in this anonymous study voluntarily.

Among the data obtained from the first part of the questionnaire, a correlation matrix was performed between the orthopedic surgeon’s confidence and the number of operations performed per year. The responses obtained in the Clinical Cases section were grouped by factors (life expectancy, metastasis location, etc.) and then compared using the Mann–Whitney U Test. Then, the responses in the Clinical Cases section were divided into groups on the basis of work experience, workplace, and the number of operations performed over a year and next compared using the Mann–Whitney U Test. The level of significance was set at *p* < 0.05. The statistics were performed using the Statistica 13.0.2 program (StatSoft Polska Co. Ltd., Kraków. Poland).

## 3. Results

### 3.1. Participant Characteristics

A total number of 104 responses were collected from members of the Polish Society of Orthopedics and Traumatology. Most of the respondents were male (*n* = 97, 93.3%). The respondents mostly indicated that they had been working in the profession for longer than 11 years (*n* = 64, 61.6%). A similar number of surgeons working in university hospitals (*n* = 43, 41.3%) to districts hospitals (*n* = 50, 48.1%) took part in the study. All the demographic results are shown in Table 1. After the correlation matrix was performed, a positive correlation (0.5262) was seen between the number of operations performed per year and the level of confidence in MBD surgeries (*p* < 0.001).

### 3.2. Diagnosis and Qualification for Surgery

The most frequently chosen decision in the case of a patient presenting with suspected MBD was: ‘Admission to the ward and qualification for further treatment’ (47.1%; *n* = 49). However, approximately 36.5% of the respondents (*n* = 38) would choose to refer the patient to an orthopedic oncology center. On the other hand, 17.3% (*n* = 18) would first refer the patient to an oncology clinic. More than half the respondents (56.7%; *n* = 59) indicated that they did not use any scales and classification when qualifying a patient with MBD for surgery. The most frequently chosen scale was the Musculoskeletal Tumor Society Scoring system (MSTS) and Mirels Classification—both 19.2% (*n* = 20) of responses. Only 3.8% of surgeons (*n* = 4) indicated that they know and use the PATHFx application to assess the patient’s prognosis and survival. On the other hand, 25% of respondents (*n* = 26) know the above-mentioned application but do not use it in their practice. Additionally, surgeons were asked how much they agreed with the statement: ‘Patients with a single lesion in bone suspected of a metastasis lesion should have a biopsy performed to exclude primary bone tumors’. The vast majority of respondents (69.2%) answered, ‘I strongly agree’ (44.2%) and ‘Agree’ (25%). Only 13.5% of the respondents did not agree with this statement. The most frequently chosen radiology examinations and the type of preferred biopsy are given in Figure 1.

This part of the questionnaire also examined the importance of some factors for orthopedists in qualifying a patient with MBD for surgery. Figure 2 shows which factors were chosen as the most important. When qualifying a patient with MBD for surgery, the respondents most often take into account such factors as the level of pain (78.8%), the number of bone metastases (84.6%), size of the lesion and the level of bone destruction (91.4%), the occurrence of pathological fracture (95.2%), impending fracture (92.3%), patient life expectancy (72.1%), type of primary tumor (72.1%), visceral metastases (65.4%), functional assessment and patient mobility (81.8%) and quality of life (80.8%). Less frequently, orthopedists in their practice take into account such factors as the plasma hemoglobin concentration (42.3%) and the time from the diagnosis of cancer to the detection of a metastatic lesion in bone (47.1%).

### 3.3. Treatment

In the cases of well-vascularized bone metastases, the surgeons’ responses were almost evenly distributed: 38.5% of the respondents did not use preoperative embolization at all or rarely; 21.2% use it only sometimes, while 30.4% use it always or often. After surgery, the respondents most often refer patients to an oncology clinic (85.6%) or an orthopedic clinic (14.4%). In clinical practice, the respondents most often use Polymethyl methacrylate (PMMA) (85.6%) to fill in bone loss after resection of a metastatic lesion. Other methods are shown in Figure 3.

### 3.4. Clinical Cases

In the choice of the method of surgical treatment in patients with MBD a clear trend was observed. The most frequently chosen methods were: IMN + Resection + PMMA (30.47%) and IMN without tumor resection (42.13%), and in third place, modular endoprosthesis (17.25%) (Figure 4A). Regarding the type of cancer, it was observed that respondents more often chose a modular endoprosthesis (megaprotheses) with tumor resection in the case of kidney cancer (20.8%) compared to breast cancer (13.9%) (*p* = 0.003) (Figure 4B). Significant differences can also be observed in the choice of treatment method with the variable “Patient’s life expectancy”. In the cases of patients with a life expectancy of more than 12 months, doctors more often chose the method: IMN + Resection + PMMA (46.5% vs. 14.7%) (*p* = 0.000923) and modular endoprosthesis with tumor resection (29.3% vs. 5.4%) (*p* = 0.000899). In patients with a life expectancy of less than 6 months, the most frequently chosen method was: IMN without tumor resection (17.8% vs. 66.7%) (*p* = 0.000923) (Figure 4C). It was shown that the respondents more often would not qualify the patient for surgery (1.3% vs. 10.1%) (*p* = 0.003471) in the case of no visible fracture on X-ray, even if there is a high risk of fractures in the near future (Impending fracture). The respondents also more often decided to refer patients to an orthopedic oncology center than in cases without presented fracture (0.6% vs. 3%) (*p* = 0.004752) (Figure 4D). On the other hand, the analysis of cases in terms of tumor localization showed that the respondents more often preferred stabilization with a plate-screw fixation device in the case of the humerus than the femur. (3.9% vs. 0.6%) (*p* = 0.047914) (Figure 4E). In question 45, in the case of a patient eligible for radical treatment, the majority of respondents would decide to remove the tumor (75%, *n* = 78), of which 44.2% (*n* = 46) would perform IMN + Resection + PMMA, while 30.8% (*n* = 32) would perform a modular endoprosthesis with resection of the lesion. Approximately 22.2% (*n* = 23) would not decide to perform the tumor resection—IMN without tumor resection (8.7%, *n* = 9); There are no indications for surgery (13.5%, *n* = 14).

Then, it was assessed whether factors such as work experience, main workplace, or the number of metastatic surgeries performed per year had an impact on the choice of treatment method among orthopedic surgeons. The respondents were divided into two groups in terms of work experience ≤10 years and ≥11 years, and it was observed that the less experienced group of orthopedic surgeons more often (47.5% vs. 39.5%) decided to perform IMN without tumor resection than the group of more experienced physicians (*p* = 0.046). In the case of the workplace, surgeons from district hospitals less frequently (13.7% vs. 23.1%) would decide to use modular endoprosthesis than surgeons from university hospitals (*p* = 0.000076). Orthopedists who performed ≥11 bone metastases surgeries per year would also more often use modular endoprosthesis (34.8% vs. 13.2%) than doctors who performed ≤10 operations per year (*p* = 0.000114) (Table 2).

## 4. Discussion

In the study, we observed a trend in the selection of treatment methods for patients with MBD, which partially coincides with the guidelines in scientific publications, where the main methods of treatment of metastases to the shaft of long bones are IMN with tumor resection, modular endoprostheses, and IMN without tumor resection [8,11,13]. However, to analyze these trends in more detail, it is important to look at how certain factors (life expectancy, type of cancer, etc.) influenced treatment choices among the respondents. The histopathological type of the primary cancer is undoubtedly important in the selection of the method of surgical treatment for bone metastases [11]. Some types of cancer, including kidney cancer, which was used in our survey, are more likely to qualify for radical surgery [11]. First, after the resection of a single metastatic renal cancer, the expected five-year survival rate is 35%; hence an aggressive approach is warranted [7]. Additionally, 25% of patients have a life expectancy of three to ten years; therefore, longer-lived fixation methods are recommended [5,7]. It is, therefore, not surprising that more frequent use of modular endoprosthesis with resection was observed in the responses in the cases of metastatic renal carcinoma. In contrast, the second type of cancer used in the survey, breast cancer, has a good response to radiotherapy [14,15]. However, it has not been shown that the wide excision of a metastatic lesion in breast cancer improves patient survival [15,16]. Nevertheless, it is worth emphasizing that in the case of pathological fractures, bone metastases from breast cancer show a relatively good tendency to heal; therefore, minimally invasive methods of osteosynthesis should be considered in these patients, especially if we are dealing with a patient with a short life expectancy [16]. This is also confirmed by our study, where in the cases of breast cancer, respondents more often decided to use less invasive methods, such as IMN without tumor resection.

The occurrence of a pathological fracture is an unfavorable prognostic factor [17]. Additionally, fracture lowers the patient’s quality of life and generates high costs for the health system [18]. Another group is the patients with a high risk of fracture in the near future—impending fracture. In the event of a pathological fracture, the recommendations refer to the need for surgical treatment if the patient’s condition allows it [7,8]. This is also confirmed by our research, where only 1.3% of the respondents would not decide to perform surgery in the case of a pathological fracture. However, in the case of patients with impending fracture, there are no clear guidelines that would indicate the need for prophylactic stabilization. On the one hand, as reported by B. Rath et al., on average, only about 10% of patients with long bone metastases will develop a pathological fracture [19]. In addition, some authors emphasize the lack of need for prophylactic stabilization, especially in the case of metastases located in the upper limb [7,11]. On the other hand, some authors emphasize the fact that pathological fracture is an unfavorable prognostic factor for patients and that fracture treatment gives worse functional results than prophylactic stabilization [8,13]. Therefore, in the case of patients with more than 8 points on the Mirels scale, with a metastatic lesion occupying more than 50% of the cortical layer and with pain that cannot be controlled with radiotherapy and anti-resorption therapy (denosumab), surgery and prophylactic stabilization should be considered [7,19]. Our study showed a higher frequency of withdrawal from surgery in the absence of a pathological fracture. It is worth noting, however, that in the case of patients with impending fracture, a greater number of surgeons would decide to refer patients to an orthopedic oncology center.

In the case of patients with long bone metastases, surgery is recommended by some authors even when the life expectancy is only 2–6 weeks [5,20]. At the same time, the very choice of the treatment method depends to a large extent on the expected survival time of the patient [8]. For example, the IMN used in the surgical treatment of bone metastases has a shelf life of up to 12–24 months [11]. Hence, if the patient’s life expectancy is greater, another method of stabilization should be considered. Additionally, the degree of invasiveness of the procedure should be considered so that the postoperative rehabilitation process itself does not take more time than the predicted life expectancy of the patient. Therefore, the accurate estimation of the survival time of patients is of great importance. The most frequently used system so far has been the division of patients into four classes described by Capanna and Campanacci [7]. It is worth mentioning, however, that in 2014, the Italian Orthopedic and Traumatology Society (SIOT) Bone Metastasis Study Group developed its own management algorithm [21]. In our study, each case was classified as grade 2 or 3 according to Capanna, and according to the above-mentioned Italian guidelines, each of our patients was eligible for surgery. In patients with a short life expectancy (<6 months), minimally invasive procedures should be considered first, in particular, an intramedullary nail without tumor resection [7,8,22]. For patients with a longer life expectancy (>12 months), the options to consider were modular endoprosthesis and intramedullary nail with tumor resection [6,19,23]. We observed similar trends in the responses to the survey.

Taking into account the location of the lesion (femur/humerus), we did not observe any significant differences. The exception was the use of plate and screw osteosynthesis, which was more often chosen for metastatic lesions located in the humerus (3.9% vs. 0.6%). This is justified because, according to the literature, stabilization with plates is appropriate in the case of metastases located on the upper limb (e.g., the humerus shaft), which is not exposed to such high loading forces as the lower limb [11]. However, it should be remembered that the proximal and distal cortex must be of good quality and that the plate should be long enough to withstand the torsional forces acting on the limb [11].

In the case of patients with a single bone lesion and the inability to exclude a primary bone tumor with a high probability, it is advisable to refer the patient to an orthopedic oncology center for a biopsy [24]. According to T.I. Wang et al., patients with primary bone tumors diagnosed and treated outside the orthopedic oncology center showed a higher rate of recurrence (42% vs. 22%) [25]. Additionally, B. Wang et al. showed that patients with bone sarcomas treated outside the oncology orthopedics center had earlier lung metastases (6 months vs. 16 months) compared to patients treated in hospitals with the highest level of reference [26]. Currently, reports indicate the benefits of treating patients with bone tumors in specialized oncological orthopedics centers [27,28]. The recommendations of the European Cancer Organization (ECCO) indicate that treatment of patients with bone tumors should be dealt with by centers that have experience of, inter alia, 50 cases a year [27]. Our study showed trends in the choice of surgical treatment depending on the number of metastatic surgeries performed per year. In the case of orthopedics centers with little experience in oncology (for example, district hospitals), surgeons working there less frequently would decide to use modular endoprosthesis with radical removal of the metastases. In addition, surgeons with work experience of ≤10 years more often decide to treat patients with MBD using IMN without tumor resection, which is less invasive and technically easier to perform than a modular endoprosthesis. It can therefore be seen that patients with bone tumors, including bone metastases, do not have equal access to all treatment methods in all hospitals. It is worth emphasizing here, however, that 100% of responses to the ‘Clinical cases’ section—referral to an orthopedic oncology center—came from doctors working in district hospitals. Therefore, it can be assumed that doctors with little experience in the oncology of the musculoskeletal system, if they have such an opportunity, refer the patient to the center with the highest level of referentiality.

As shown by J.C. Weeks et al., physicians tend to be overly optimistic about patient survival [29]. “Excessive optimism” may lead to an incorrect choice of treatment methods and may result in patient dissatisfaction with the treatment administered [5]. Therefore, the use of objective scales to assess the patient’s survival is, in our opinion, absolutely justified and helps to select the appropriate type of treatment. Our research shows, however, that among the respondents, more than half of the surgeons (57%) do not use any scales. It is also worth mentioning that only about 4% of respondents use the PATHFx application proposed by the International Bone Metastasis Registry and validated in 2020 to assess the expected survival time of patients [30]. Although the scales are not ideal tools, in the case of surgeons with little experience in the oncology of the musculoskeletal system, the use of these can help in the appropriate selection of treatment methods and direct the patient further to ensure the best possible chance for the appropriate treatment. It is also worth emphasizing that only 20% of patients with bone metastases qualify for surgery [5]. Hence, determining the factors considered by orthopedists when qualifying for surgery is of great importance. In our study, respondents consider factors that are most often mentioned as important in publications when qualifying patients with MBD for surgery [8,19].

A biopsy in the case of suspected bone metastases is indicated primarily in tumors of unclear origin (differentiation from bone sarcomas and benign, locally aggressive lesions) [31]. Moreover, in our study, 69% of the respondents believe that in the case of a single bone lesion, even in patients with a history of oncology, a biopsy is indicated. The choice of the biopsy method depends on the orthopedic center and the experience of the surgeons; however, the most frequently chosen methods in the case of bone tumors are surgical (open) biopsy and core-needle biopsy [32]. More than half of the respondents in our study (55%) prefer a surgical biopsy, while approximately 20% would choose a core-needle biopsy or a trepanobiopsy. Among radiological tests, the first choice for bone tumors is X-ray (assessment of the lesion—osteosclerotic\osteolytic) [8]. Although MRI, bone scan, and PET-CT are useful tests to obtain more accurate imaging of lesions in the bone and for differential diagnosis, they are not always available to orthopedic surgeons [33]. This is confirmed by our study, where X-ray was chosen by 86% of respondents, MRI by 70%, bone scan by 52%, and PET-CT by 23%. This is also evidenced by the high percentage of computed tomography (CT) performed in the case of suspected MBD—91% of respondents. In the case of small orthopedic centers, in the absence of tests such as MRI or bone scans, surgeons may more often decide to examine patients with MBD using a CT. Although CT allows for the determination of bone tissue destruction and allows the assessment of the risk of fracture, only MRI allows for a more accurate analysis of the soft tissue infiltration by the tumor and a better estimation of the biopsy site [19,33].

According to J. Ma et al., preoperative embolization is indicated in the case of well-vascularized tumors in order to, e.g., limit intraoperative bleeding [34]. However, this method is not used by all orthopedic centers [11]. Our study also shows this, where 23% of respondents declare that they always use embolization for well-vascularized secondary bone tumors. On the one hand, it may result from the lack of access to the interventional radiology clinic. On the other hand, according to Geraets et al., there is no hard evidence that would confirm the efficacy of preoperative embolization in reducing blood loss in a patient with MBD during surgery [35]. This is not the case with postoperative radiotherapy, which is widely used to reduce the risk of recurrence and subsequent loosening of the implant, loss of limb function, and aggravation of pain [36]. Therefore, in our opinion, a referral to an oncology clinic to determine a further therapeutic path is justified. The participants of the study are of a similar view, where 84% of doctors refer a patient with MBD after surgery to an oncology clinic. Filling in bone defects in the case of bone metastases is often a great challenge for surgeons. Polymethyl methacrylate (PMMA) is most often used in orthopedic clinical practice due to its easy availability and low price. In addition, it increases structural stability, allows the patient with MBD to return to functionality faster, and strengthens local control after tumor curettage. Disadvantages include the longer operation time, the risk of impaired wound healing, and local bleeding [11,37]. Cementing requires the use of low-viscosity PMMA which is injected under minimal pressure. It is also important to hydrate the patient properly to reduce the risk of fatty embolism [38]. It is, therefore, not surprising that approximately 86% of the respondents choose PMMA to fill in bone defects during metastatic surgery.

One of the limitations of our work is the number of participants. The number of 104 respondents does not seem to be a sufficiently representative group for the population of orthopedic surgeons in Poland. However, if we consider that in Poland, only some centers deal with the oncology of the musculoskeletal system and that most smaller hospitals refer patients with MBD to an orthopedic oncology clinic, it is unsurprising that the number of responses to the questionnaire is low. In addition, it should be emphasized that despite the relatively small group of respondents, we managed to achieve a number of respondents with different levels of experience and different workplaces, which allowed us to observe trends in the choice of treatment methods. Another limitation of our work is the lack of validation of the questionnaire. It is worth noting, however, that the survey was based on a scientific study by M.Steensm et al. [22]. We are also aware of the difficulties that the presented cases could have caused our respondents due to the limited availability of data on a specific patient. In our study, however, we wanted to check how the given factors affect the choice of treatment method by surgeons, and although some respondents could conduct treatment differently in reality than in the survey, the observed trend indicates that most doctors understood the presented cases well.

## 5. Conclusions

The trends in the surgical treatment of metastases to long bones are consistent with the data available in the literature, with the most frequently chosen methods IMN and modular endoprosthesis (megaprotheses). Length and type of professional experience, place of work, and the number of metastasis surgeries performed during a year may influence the choice of treatment method in patients with MBD. Elements that could improve the quality of treatment of patients with MBD are the more frequent use of objective scales by orthopedists when qualifying patients for surgery, improving access to radiological examinations (MRI, bone scans), and establishing specialized orthopedic oncology centers for patients with MBD.

## Figures and Tables

**Figure 1 jcm-11-04284-f001:**
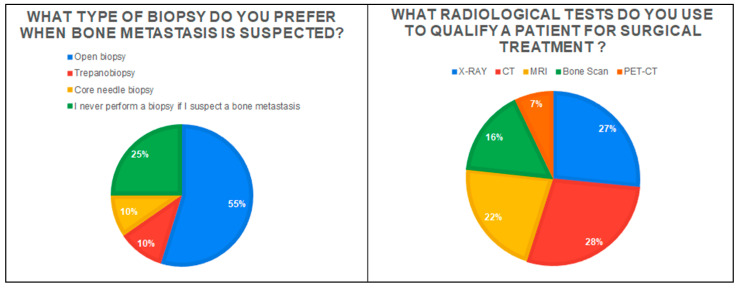
The graph on the left shows the type of biopsy preferred for patients with metastatic bone disease. The chart on the right shows the radiology examinations most frequently chosen by respondents in the diagnostic process of patients.

**Figure 2 jcm-11-04284-f002:**
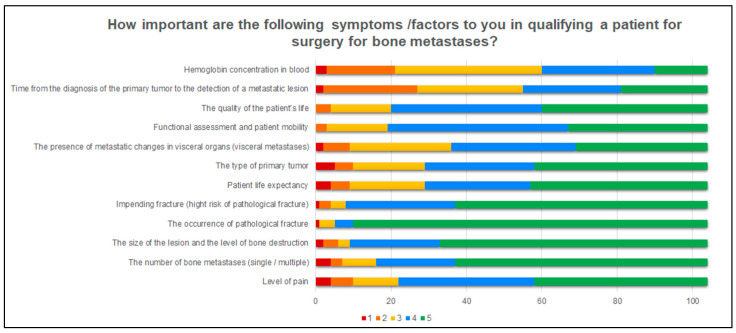
A graphic presentation how important the specifics factors are for the respondents when qualifying a patient for surgery. The answers were divided according to the Likert scale: 1-Completely unimportant, 2, Slightly unimportant; 3, Slightly important; 4, Moderately important; 5, Very important.

**Figure 3 jcm-11-04284-f003:**
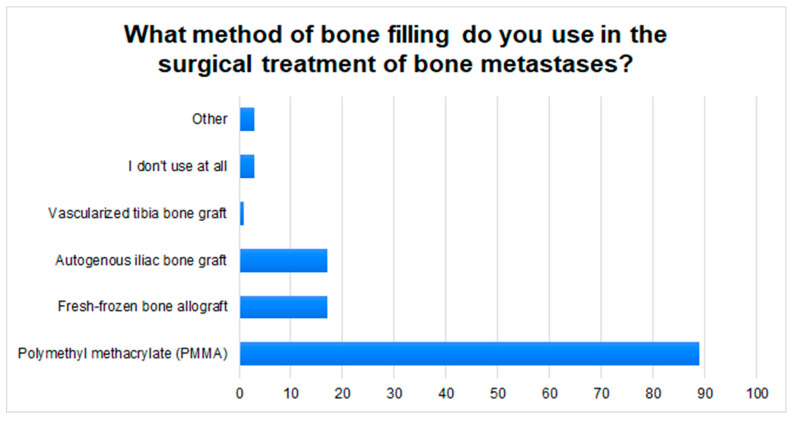
The graph shows the most commonly used methods of filling bone defects in patients with MBD, in the respondents’ clinical practice.

**Figure 4 jcm-11-04284-f004:**
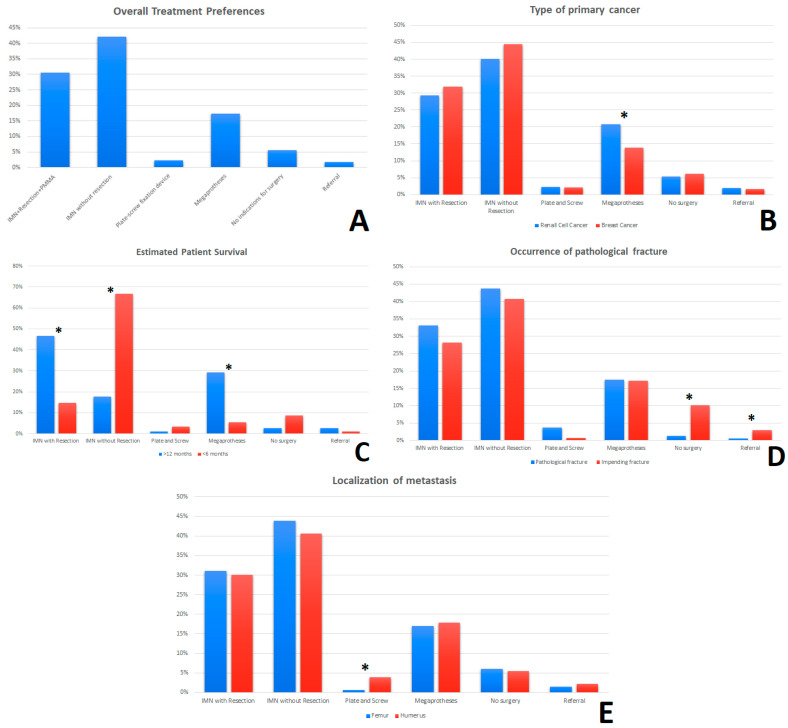
The survey responses in the “Clinical cases” section were combined with the frequency of selecting the surgery method of treatment and recorded as percentages (**A**). The percentages of the breakdown of overall responses by—type of primary cancer (**B**), estimated time of survival (**C**), occurrence of pathological fracture (**D**) and anatomical localization of metastases (femur/humerus) (**E**) are shown. * An asterisk indicates a statistically significant difference (*p* < 0.05) for the comparisons between groups (Mann-Whitney U test). Abbreviation: IMN—Intramedullary nail; PMMA—Polymethyl methacrylate.

**Table 1 jcm-11-04284-t001:** Distribution of survey responses in the first section—Participant characteristic. In the ‘Main field of interests’ column, respondents could choose up to 3 domains (multiple choice question).

	Percentage (*n* = 104)
**Gender**	
- Female	6.7%
- Male	93.3%
**Work experience** [years]	
- 0–5	16.3%
- 6–10	22.1%
- 11–20	38.5%
- >20	23.1%
**Main workplace**	
- University Hospital	41.3%
- District Hospital	48.1%
- Private medical practice	4.8%
**Number of performed surgeries for bone metastases** [number per year]	
- 0–5	59.6%
- 6–10	20.2%
- 11–20	6.7%
- >20	13.5%
**Main field of interests**	
- General orthopedics	58.7%
- Traumatology	66.3%
- Musculoskeletal oncology	22.1%
- Joints arthroplasty	59.6%
- Arthroscopy and minimal invasive orthopedics	34.6%
- Spine surgery	12.5%
- Hand surgery	12.5%
- Pediatric orthopedics	4.8%
- Hand surgery	20.2%
- Pelvic surgery	1%
**How confident do you feel when performing surgery on metastatic bone tumors?**	
- Definitely unconfident	18.3%
- Rather unconfident	18.3%
- Neutral	33.7%
- Rather confident	23.1%
- Definitely confident	6.7%

**Table 2 jcm-11-04284-t002:** Comparison of the choice of surgical treatment method in patients with MBD among surveyed surgeons. Significantly statistical differences are marked in red—*p* value < 0.05 (Mann–Whitney U test). Abbreviation: IMN—Intramedullary nail; PMMA—Polymethyl methacrylate.

	IMN+ Resection+ PMMA	IMN without Tumor Resection	Plate-Screw Fixation Device	Modular Endoprosthesis with Tumor Resection	No Indications for Surgical Treatment
**Work experience** [years]					
≤10	29.8%	47.5%	2.8%	13.2%	6.7%
≥11	32.5%	39.5%	1.9%	21.3%	4.9%
**Workplace**					
University Hospital	29.3%	37.7%	3.64%	23.1%	6.3%
District Hospital	32.1%	47.4%	1.4%	13.7%	5.3%
**Number of surgeries** [number per year]					
≤10	33.2%	45.2%	1.8%	13.2%	6.6%
≥11	24.4%	35.1%	3.6%	34.8%	2.1%

## Data Availability

Not applicable.

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
