# Peer review of "Trends in Diagnosis and Surgical Treatment of Bone Metastases among Orthopedic Surgeons"

_jcm, 2022, doi:10.3390/jcm11154284_

Round 1
Reviewer 1 Report
Thanks very much to the editor for giving me this opportunity to review this article!This is a very meaningful and interesting article. The idea in this manuscript is very novel. This study aimed to figure out what factors incfulence the choice of the treatment. This is a carefully done study and the findings are of considerable interest. My detailed comments are as follows:a. what is basis for the questionnaire. b. Plesea expain the differences between the questionaire and others.
Author Response
Dear Reviewer,
Thank you very much for reviewing our manuscript. Thank you also for the attached comments, to which I am sending the respond below:
a) When creating the questionnaire, we took into account the data available from the literature and the current guidelines in the treatment of metastases to the diaphysis of long bones (Reference in the manuscript). Our goal was to check what methods of diagnosis, qualification and treatment are currently used by orthopedists in Poland. The part concerning clinical cases was created on the basis of this publication (Steensma, M.; Healey, J.H. Trends in the Surgical Treatment of Pathologic Proximal Femur Fractures among Musculoskeletal Tumor Society Members. Clin. Orthop. Relat. Res. 2013, 471, 2000–2006, doi:10.1007/S11999-012-2724-6.). The location of the metastatic lesion (long bone diaphysis) was changed in order to assess the factors influencing the choice of radical surgical treatment methods (megaprotheses) compared to non-radical methods (intramedullary devices without tumor resection).
b) To our best knowledge, this is the first questionnaire, which examine what methods of diagnosis and qualification for surgical treatment are preferred by orthopedic surgeons. Moreover, it is the first questionnaire that examines which surgical methods are preferred by orthopedists in cases with metastases to long bone diaphysis. The advantage of our questionnaire compared to the previously published ones is that it examines both the process before surgery and the factors influencing surgical treatment.
Thank you again for review and any comments. We hope that the submitted answer will meet your expectations. Otherwise, we remain at your disposal and we will gladly answer any doubts.
With kind regards,
Dawid Ciechanowicz
Reviewer 2 Report
Dear authors,
thank you for submitting this nice piece of work.
In the manuscript you describe and analyze the present diagnostic and surgical treatment preferences of bone metastases among polish orthopedists. Therefore you designed a questionnaire, which has been completed by 104 persons. In the end you analyze whether there is a difference in the treatment regimen depending on the level of experience and you give some suggestions how the treatment of patients with BMD could be improved in the future. As you write, this is a very important topic as the incidence of cancer is still rising. The decision whether to operate or not, or how "aggressive" the surgical treatment should be, must be determined by facts.
As you write, the weakness of the study is the quite low number of respondents, but you analyze that regarding experience and working place its a representative cohort of the polish orthopedic society. The manuscript is easy to read and to understand. The references used are meaningful. There are only a low number of small mistakes to rectify:
- Table 1: "Work esperancie" ->work experience
- Table 1: there is a misalignment between the topic and the percentage (look at the "main field of interests" section)
- Line 131: 8 -> brackets are missing
- Line 166 / 168: "U Manna-Whitney Test" -> Mann-Whitney U Test
- Lines 219 ff: In the cases of well-vascularized.... did not use this method at all.... -> which method ? Embolization? It is not mentioned before. Please clarify.
-Line 296: 7 -> brackets missing
- Line 306: 16 ->brackets missing
- line 318: 19 -> brackets missing
Author Response
Dear Reviewer,
Thank you for giving us the opportunity to submit a revised draft of the manuscript. We appreciate the time and effort that you dedicated to providing feedback on our manuscript and are grateful for the insightful comments on and valuable improvements to our paper.
We attach the respond to the comments below:
Thank you for listing any mistakes in the manuscript text. All corrections were made and saved in the attached version of the manuscript.
In case of further comments, we remain at your disposal and we will be happy to answer any doubts.
With kind regards,
Dawid Ciechanowicz
